# An Assessment of Health Outcomes and Methylmercury Exposure in Munduruku Indigenous Women of Childbearing Age and Their Children under 2 Years Old

**DOI:** 10.3390/ijerph181910091

**Published:** 2021-09-25

**Authors:** Joeseph William Kempton, André Reynaldo Santos Périssé, Cristina Barroso Hofer, Ana Claudia Santiago de Vasconcellos, Paulo Victor de Sousa Viana, Marcelo de Oliveira Lima, Iracina Maura de Jesus, Sandra de Souza Hacon, Paulo Cesar Basta

**Affiliations:** 1Faculty of Medicine, St Mary’s Hospital, Imperial College London, London W2 1PG, UK; joe.kempton@nhs.net; 2Departamento de Endemias Samuel Pessoa, Escola Nacional de Saúde Pública, Fundação Oswaldo Cruz (ENSP/Fiocruz), Rua Leopoldo Bulhões, 1480, Manguinhos, Rio de Janeiro 21041-210, Brazil; aperisse41@gmail.com (A.R.S.P.); sandrahacon@gmail.com (S.d.S.H.); 3Instituto de Pediatria e Puericultura Martagão Gesteira, Faculdade de Medicina, Universidade Federal do Rio de Janeiro (UFRJ), Rua Bruno Lobo, 50, Cidade Universitária, Rio de Janeiro 21941-912, Brazil; cbhofer@hucff.ufrj.br; 4Laboratório de Educação Profissional em Vigilância em Saúde, Escola Politécnica de Saúde Joaquim Venân-cio, Fundação Oswaldo Cruz (EPSJV/Fiocruz), Av. Brasil, 4365, Manguinhos, Rio de Janeiro 21040-900, Brazil; anacsvasconcellos@gmail.com; 5Centro de Referência Professor Hélio Fraga, Escola Nacional de Saúde Pública, Fundação Oswaldo Cruz (CRPHF/ENSP/Fiocruz), Estrada de Curicica, 2000, Curicica, Rio de Janeiro 22780-195, Brazil; paulovictorsviana@gmail.com; 6Seção de Meio Ambiente, Instituto Evandro Chagas, Secretaria de Vigilância em Saúde, Ministério da Saúde (SEAMB/IEC/SVS/MS), Rodovia BR-316 km 7 s/n, Levilândia 67030-000, Brazil; marcelolima@iec.gov.br (M.d.O.L.); iracinajesus@iec.gov.br (I.M.d.J.)

**Keywords:** environmental health, indigenous people, Amazon, 1000 days, childbearing women, nutrition, vaccine coverage, mercury exposure

## Abstract

In line with the 1000-day initiative and the Sustainable Development Goals (SDG) 2 and 3, we present a cross-sectional analysis of maternal health, infant nutrition, and methylmercury exposure within hard-to-reach indigenous communities in the state of Pará, Brazilian Amazon. We collected data from all women of childbearing age (i.e., 12–49) and their infants under two years old in three Munduruku communities (*Sawré Muybu*, *Sawré Aboy*, and *Poxo Muybu*) along the Tapajos River. We explored health outcomes through interviews, vaccine coverage and clinical assessment, and determined baseline hair methylmercury (H-Hg) levels. Hemoglobin, infant growth (Anthropometric Z scores) and neurodevelopment tests results were collected. We found that 62% of women of childbearing age exceeded the reference limit of 6.0 μg/g H-Hg (median = 7.115, IQR = 4.678), with the worst affected community (*Sawré Aboy*) registering an average H-Hg concentration of 12.67 μg/g. Half of infants aged under 24 months presented with anemia. Three of 16 (18.8%) infants presented H-Hg levels above 6.0 µg/g (median: 3.88; IQR = 3.05). Four of the 16 infants were found to be stunted and 38% of women overweight, evidencing possible nutritional transition. No infant presented with appropriate vaccination coverage for their age. These communities presented with an estimated Infant Mortality Rate (IMR) of 86.7/1000 live births. The highest H-Hg level (19.6 µg/g) was recorded in an 11-month-old girl who was found to have gross motor delay and anemia. This already vulnerable indigenous Munduruku community presents with undernutrition and a high prevalence of chronic methylmercury exposure in women of childbearing age. This dual public health crisis in the context of wider health inequalities has the potential to compromise the development, health and survival of the developing fetus and infant in the first two critical years of life. We encourage culturally sensitive intervention and further research to focus efforts.

## 1. Introduction

Nutrition during the first 1000 days from a woman’s pregnancy to the child’s second birthday is a critical stage in determining the child’s prospects of both growth and learning [1]. This period represents a stage of life with great potential for human development, but equally, of enormous vulnerability to the harm associated with factors such as a lack of essential elements (e.g., vitamins, minerals, fatty acids), infections caused by microorganisms (e.g., diarrhea, pneumonia), and exposure to toxic substances such as medicines (e.g., antibiotics) and environmental pollutants (e.g., pesticides, lead, mercury) [2,3,4]. An understanding of the importance of this period drives government policy and intervention worldwide with the 1000 Day Initiative outlining how poor nutrition during this period can keep families and communities trapped in poverty [5,6,7]. In the same vein, the United Nations adopted the Sustainable Development Goals (SGDs) in 2015 as a planetary alert to end poverty by 2030. Goals 2 and 3 set out to ensure access to safe and nutritious food all year round and reduce illness from hazardous chemicals and water contamination [8,9].

Indigenous communities living in remote areas of the Brazilian Amazon have limited access to varied food sources, relying heavily on fish from nearby rivers for their supply of protein [10,11,12,13,14,15]. Besides this food insecurity, these communities also face risk factors such as gastrointestinal infections, respiratory diseases, poor sanitation, and alarming rates of anemia [16,17,18]. This condition of vulnerability imposed on the indigenous people in Brazil is known to negatively impact children’s cognitive development, physical growth and immunity [16,19,20,21]. In fact, indigenous children are twice as likely to be affected than other children of the same age [17]. Similarly, studies have also found deficits in height-for-age, weight-for-age, and weight-for-height [18,19,22], as well as increased rates of infant mortality [23,24].

Another important health risk factor faced by indigenous populations living in the Amazon is mercury exposure, which has been described in several scientific papers [11,12,13,14,15]. Mercury is an extremely toxic heavy metal widely used in artisanal gold mines, installed in practically the entire Amazon region [10,25,26,27]. Mercury used in mines is methylated by bacteria at the bottom of rivers and transformed into its most dangerous form to health, methylmercury (MeHg) [28]. This organomercurial form is a potent neurotoxin and biomagnifies along the entire aquatic trophic chain, contaminating fish, which are often the main source of protein in the diet of many indigenous and riverine communities in the Amazon [29,30]. Factors such as deforestation, burning and the construction of dams for hydroelectric plants intensify the formation of methylmercury and increase human exposure to this toxic substance [10]. The toxicological effects of chronic human exposure to methylmercury through the consumption of fish are pernicious with symptoms including cognitive, visual, motor, somatosensory, attention and emotional deficits as well as immunotoxic or immunomodulatory effects [29,30,31,32,33,34,35,36,37].

It is important to emphasize that neurotoxic effects related to methylmercury exposure are most serious during the prenatal period, since the fetal brain is particularly sensitive to the action of any neurotoxic substances [38,39,40,41]. When methylmercury contaminated fish is consumed, this mercurial form is almost completely absorbed by the gastrointestinal tract and rapidly circulates throughout the body, ultimately reaching its principal target, the central nervous system. In pregnant women, this toxicokinetic pattern is repeated, with methylmercury then crossing the placental barrier and entering the fetal circulatory system [38,39,40]. Some studies indicate that cord blood corresponds to maternal blood at a ratio of 1.7, leaving the developing nervous system exposed to amplified neurotoxic effects [40,41,42]. Exposure to a neurotoxin at such a critical period of development has been shown to lead to physical and mental developmental delay, altered muscle tone and reduced neurological test scores, even after exposure to levels of methylmercury that had little effect on the mother [38,39,40,41,42]. Mercury exposed infants specifically have been found to have reduced cognitive abilities [43].

There is little (or no) consensus amongst agencies in their recommendations for the safe ingestion of methylmercury. In 1972, the Joint Food and Additive Organization/World Health Organization (FAO/WHO) Expert Committee on Food Additives (JECFA) established a provisional tolerable weekly intake (PTWI) for methylmercury equal to 3.3 µg/kg bw/week, drawing on health endpoints from poisoning episodes in Minamata and Niigata in the 1950s [44]. Years later, in 1997, the United States Environmental Protection Agency (U.S.EPA) proposed a Reference Dose (RfD) for methylmercury of 0.1 µg/kg bw/day based on the intoxication tragedy in Iraq, which was revised and maintained after the Faroes Island cohort study [45]. More recently, in 2003, the JEFCA established PTWI for most vulnerable groups, as women of childbearing age and children, of 1.6 µg/kg bw/week, and for adults in general of 3.2 µg/kg bw/week [46].

These safe intake doses proposed by different agencies correlate with mercury exposure biomarkers, such as blood and hair, and the levels detected in these matrices can be used as a risk exposure indicator. For example, in 1989, the JEFCA proposed a PTWI of 0.3 mg of total mercury per person of which cannot surpass 0.2 mg of methylmercury [47]. This PTWI was converted to hair mercury levels of 6.0 µg/g for identifying individuals with high mercury exposure in the New Zealand cohort [48]. Many studies developed in the Amazon region used this mercury level as a reference for the appearance of health effects [49,50,51], including research with indigenous peoples with a history of or suspected exposure to mercury used in illegal gold mining [13,52].

The Munduruku indigenous people living in the Middle Tapajós Region have been subjected to invasions of their territories by gold miners for decades. However, in recent years, the process of invasion has intensified and caused enormous concern for local communities. With this in mind, the *Pariri* Indigenous Association, representing the Munduruku living in the Middle Tapajós, asked the Oswaldo Cruz Foundation to carry out a survey to investigate the extent of mercury contamination amongst the Munduruku indigenous people.

In a recent study, Vasconcellos et al. [12] found that the fish consumed by three Munduruku villages in the Middle Tapajós region showed mercury levels above the limits for commercialization established by FAO/WHO [53] and showed the methylmercury intake in these communities to be several times higher than the safety doses proposed by U.S.EPA [45] and FAO/WHO [46], including the most vulnerable groups of women of childbearing age and children. This study indicates that these communities ingest methylmercury daily at levels that can cause illness and highlights the importance of further research into mercury exposure and its health effects amongst indigenous people.

Thus, in line with the *Pariri* Indigenous Association’s request, the 1000 Day initiative and UN Development Goals 2 and 3, the objective of this study was to explore the nutritional status, health outcomes and baseline methylmercury levels of all Munduruku indigenous infants and women of childbearing age in these communities with a focus on mothers who had given birth within the last two years.

## 2. Materials and Methods

### 2.1. Study Area and Population

The study was conducted in three Munduruku communities located in the *Sawré Muybu* Indigenous Land (IL) in the Tapajós River Basin, located in the municipalities of Itaituba and Trairão, in the state of Pará, Brazilian Amazon. In total, *Sawré Muybu* IL covers 178,173 hectares, with a population of approximately 800 indigenous people divided between eight villages. The people of these villages are largely subsistence farmers and fisherman, with a small number of teachers and miners [52]. The *Sawré Muybu* IL is of particular historical significance to the Munduruku community, adding further significance to the survival of Munduruku communities in this area [54].

### 2.2. Study Design

Over 10 days from 29 October to 9 November 2019, a cross-sectional study was carried out in three villages selected from *Sawré Muybu* IL (*Sawré Muybu*, *Poxo Muybu*, and *Sawré Aboy*), following the direct request from the *Pariri* Indigenous Association to the Oswaldo Cruz Foundation (Fiocruz). In the designated villages, we conducted a population census, and all residents were invited to participate in the study. There was no refusal and, therefore, no probabilistic sampling methods were used to include participants.

### 2.3. Data Collection

#### 2.3.1. Interviews

Interviews exploring mothers’ obstetric history and household information were conducted based on a data collection instrument prepared especially for this study. The questions were broadened to explore dietary patterns, information on breastfeeding practices, and pre- and post-natal complications (including miscarriages or infant deaths). In order to analyze vaccination coverage, we revised the health booklets of all indigenous infants enrolled in the study. Based on the data recorded in the child’s health booklet, it was possible to check the vaccine records, with dates and corresponding doses, and compare the records with the current national immunization program schedule for the population studied [55].

Each participant was given a unique code for identification and the questionnaire was delivered with the support of Indigenous Health Agents who work in the communities and/or with the support of the local Indigenous leaders, such as chiefs and teachers.

Responses were recorded on electronic forms with the aid of portable electronic devices (tablets), and there was no use of paper forms. After conducting home visits and interviews, families were invited to participate in a standardized clinical (i.e., neurological, and nutritional status) and laboratory evaluation (i.e., mercury analysis and hemoglobin dosage), described in the next section.

Within indigenous communities, there is a younger observational age for commencing sexual initiation and for starting families; with many already married, recent studies exploring maternal health among indigenous groups have used 10 and 14 years of age as their initial starting age [56,57]. However, for this investigation we included women between 12 and 49 years of age.

#### 2.3.2. Mercury Analysis

Hair samples were collected from all participants, removed close to the scalp in the occipital region with the aid of stainless-steel dissection scissors. The samples were stored in paper envelopes, individually identified, and sent for analysis of total mercury levels (THg) in the Toxicology Laboratory, in the Environment Section of the Evandro Chagas Institute (IEC), in Belém (Pará), Brazil. The entire methodology for determining total mercury levels can be found in Basta et al. [52], including the quality control protocol used.

We chose hair samples as methylmercury exposure biomarkers because the principal source of mercury exposure in the studied population is the intake of mercury contaminated fish and from 90% to 95% of mercury forms observed in the fish muscle samples is methylmercury [58,59,60]. Besides that, almost all mercury present in hair samples is in the methylmercury form, allowing us to assume that all mercury detected in the hair is compounded by methylmercury [61,62,63].

In addition, we also assume that the methylmercury concentration found in the mother’s hair postnatally (i.e., at the time of collection) can be used as an estimate of pre-natal exposure, during pregnancy. This assumption is because, in these communities, diet and environmental exposure patterns are consistent and relatively stable [12].

#### 2.3.3. Nutritional Status

All participants had their weight and height/length measured during the field visit using a vertical anthropometer or stadiometer from Alturexata^®^ (with adapter for infantometer and precision of 0.1 cm—length measurement) (Alturexata^®^, Belo Horizonte, Minas Gerais, Brazil) and portable digital scale from Seca^®^ (SECA^®^, model 770, Vogel & Halke, Hamburg, Germany), with a maximum capacity of 150 kg and accuracy of 0.1 kg. The measures of weight and height/length of children under 2 years old were transformed into Z-scores (adjusted for sex and age), according to the reference population of the World Health Organization (WHO), with those scoring below −2 classed as stunted [64].

Given the isolated circumstances of the fieldwork, a measurement of capillary hemoglobin was assessed using the HemoCue^®^ device (HemoCue^®^, model HB 301-System, Angelholm, Sweden) to give a point of care result, without the need to collect and store venous blood samples. Anemia was diagnosed using the WHO guidance on Anemia with those aged 6–24 months and measuring <11.0 g/dL being classed as anemic. Women of childbearing age were classed as anemic at <12.0 g/dL, however this does not signify anemia during pregnancy, which is of a lower threshold and takes dilutional activity into account.

#### 2.3.4. Neurological Status

An assessment of neurodevelopment was carried out using the Denver II developmental screening test in children aged 0 to 2 years [65]. The Denver II test assesses and identifies children at risk for developmental delay, but it is not intended to measure the intelligence quotient (IQ) and is not designed to diagnose learning or emotional disorders. The test consists of 125 items divided into 4 areas: (i) personal-social (25 items): socialization inside and outside the family environment; (ii) fine motor skills (29 items): hand-eye coordination, manipulation of small objects; (iii) language (39 items): sound production, ability to recognize, understand and use language; iv) gross motor skills (32 items): body motor control, sitting, walking, jumping and the other movements performed by the wide musculature.

### 2.4. Statistical Analysis

To commence the analyses, we performed a detailed description of the studied population, including children under two years of age and women of childbearing age, according to health parameters and hair mercury levels. The methylmercury concentration used as a reference level in women of childbearing age and their infants was 6.0 µg/g in hair samples (H-Hg) to identify individuals with high mercury exposure.

In the first stage of analysis, given the small sample size, we used the Wilcoxon signed-rank test to compare infants and women of child-bearing age to the reference limit. We then wished to examine whether levels of methylmercury in the hair of the groups of individuals we had tested differed depending on the village. Due to the small sample size and unequal variances between the groups, the Kruskal Wallis test was selected as a robust method of establishing whether any observed differences in the levels between groups were statistically relevant, with post-hoc analyses using the Mann-Whitney U test.

The next phase of the analysis focused on the relationship between maternal H-Hg and infant H-Hg levels, which was assessed using a Spearman correlation. Further descriptive analyses are given to review infant’s Denver II developmental screening test scores and the relationship between H-Hg level. Prevalence of infant stunting and infant anemia is explored in relation to Z-scores of –2 and below and <11.0g/dL, respectively.

As a proxy of the Infant Mortality Rate (IMR), we used data extracted from the household interviews where the number of infant deaths was recorded (in the numerator) in contrast to the number of live births (in the denominator), considering each mother’s response. The obtained value was multiplied by 1000.

The data were analyzed using the SciPy Open-Source library in Python, and figures were produced using the Seaborn visualization library. Throughout all statistical analyses, we set *α* < 0.05.

## 3. Results

### 3.1. Description of the Studied Population

In total, we examined 15 women and their 16 infants under two years old. The average age of the women was 22 years old (S.D. = 7.8; range from 16 to 45), while for children the average age was 10.4 months of age (S.D. = 5.3; range from 4.7 to 21.0) (Table 1). In the *Poxo Muybu* village, we examined five women and five children (three boys and two girls), between 5 to 21 months. In the *Sawré Aboy*, we evaluated four women and five children (one boy and four girls), between 5 to 23 months. Finally, in the *Sawré A*boy, we assessed six women and six children (two boys and four girls), between 5 to 11 months. Only one woman from *Sawré Aboy* had two children: one girl was 5 months, and one boy was 21 months of age (Table 1).

No infant presented with appropriate vaccination coverage for their age with health booklets of all 16 infants showing missing routine vaccines.

Three boys (7, 9, and 10 months) were missing the pentavalent vaccine that includes protection against Diphtheria, Tetanus, Pertussis, Hepatitis B, and *Haemophilus influenza* B. One girl (9 months) was missing the pentavalent and Polio vaccines. One girl (11 months) was missing the Influenza (second dose) and Meningitis vaccines. One girl (5 months) was missing the pentavalent, Meningitis, and Pneumococcus vaccines. One boy (21 months) was missing the pentavalent, Meningitis, and Influenza (second dose) vaccines. One girl (18 months) was missing the Meningitis, Hepatitis B, and Pneumococcus vaccines. One girl (7 months) was missing the pentavalent, Pneumococcus, Meningitis, Polio, and Rotavirus vaccines. One girl (4 months) missed the pentavalent, Pneumococcus and Meningitis vaccines as well as all routine vaccines that should be offered at 2, 3, and 4 months. One girl (5 months) was missing the pentavalent, Pneumococcus, Meningitis, and Rotavirus vaccines. One girl (11 months) was missing the pentavalent, Pneumococcus, Meningitis, and yellow fever vaccines, as well as all routine vaccines that should be offered from 4 months. One girl (5 months) was missing the pentavalent, Pneumococcus, Meningitis, Hepatitis B and BCG vaccines, as well as all routine vaccines that should be offered at 3 and 4 months. One boy (8 months) was missing the pentavalent, Pneumococcus, Meningitis, Rotavirus, Polio, and BCG vaccines. One girl (20 months) was missing the pentavalent, Pneumococcus, yellow fever, Influenza (second dose), and triple viral (measles, mumps, and rubella) vaccines. Finally, a boy (9 months) missed all vaccines indicated at his age.

### 3.2. Interviews

Most women described their main occupation as supporting the home, with 14 women also working in agriculture alongside domestic activities. Two women were teachers. No women described themselves as miners (also called *garimpeiros* in Portuguese). Male partners described themselves as fisherman and farmers, with five involved in mining activities.

All women described eating a variety of herbivorous and piscivorous fish frequently with *Aracu* (Herbivorous), *Barbado* (Piscivorous), *Curimata* (Detritivorous) and *Piranha* (Piscivorous) being the most common, a median frequency of three times per week was established. Nuts were eaten in varied amounts during the rainy season by all families. Hunted meat was eaten in all communities with the most common being tapir and tortoise. Fruits were eaten with banana, pineapple, and *açaí* most mentioned, access to these fruits and frequency of consumption was not established. Average income gave a monthly figure of US$242 with a range (US$0–US$1250), median US$175.

All infants aged six months and under were exclusively breastfed. Following this, a weaning process began with most infants starting to consume alternative foods, including small amounts of hunted meat and fish in the form of fish soup. Fish soup consumption varied from once per week to three times per day, with the majority of infants consuming fish soup alongside breastfeeding three times per week.

There were 53 women of childbearing age within our cohort with 35 women having given birth at least once. Five women were pregnant at the time of analysis. The number of completed pregnancies ranged from 1 to 14, with an average parity of four. Five women (9%) described a history of spontaneous abortion whilst 11 women (21%) described having at least one infant that had passed away, with a total of 13 infant deaths (Table 2). Considering the 150 live births accounted for by all these women, this puts an Infant Mortality Rate at 86.7 infant deaths per 1000 births. Importantly, this estimated IMR remained consistent within each age category assessed with ages 12 to 27 and 28 to 49 showing an estimate of 93 and 84/1000; respectively.

### 3.3. Nutritional Status Evaluation

Of the 16 infants analyzed, four (25%) were found to be moderately to severely stunted, see Figure 1. The median of height for age was −1.13 (IQR = −1.87, −1.13, and −0.24). Only one infant was found to be moderately-severely underweight. The median of weight for age was −0.58, (IQR = −1.57, −0.45, and 0.24) (Table 1).

Of infants 6–24 months (*n* = 12), six (50%) were found be anemic. The median of the hemoglobin levels was 10.9 g/dL (IQR = 10.3, 10.9, and 11.8). *Sawré Muybu* village was the worst affected village, where the median was 10.5 g/dL (IQR = 9.6, 10.4, and 11.4). Of the 15 mothers to infants, four (27%) were anemic (median = 12.9 g/dL, IQR = 11.9, 12.6, and 13.8). Of all women of childbearing age, data were collected for 52 (98%) women; eight (15.4%) were found to be anemic. The median of the hemoglobin levels was 13.3 g/dL (IQR = 12.3, 13.4, and 13.9). Of the five current pregnant women, one lacked data whilst the remaining women were not anemic (median = 13.1 g/dL).

In relation to weaning and with the data available, of infants described as eating foods other than breast milk daily (*n* = 3), no infant registered as anemic. Whilst amongst the infants registering as anemic, four out of four of the infants were recorded as weaning with alternative foods on a weekly basis.

While average BMI scores for women of childbearing age fell within the healthy range (*n* = 52; median = 23.0; IQR = 19.9, 22.0, and 25.8). In contrast, 20 women (38%) were found to be overweight (median = 26.8; IQR = 25.6, 26.3, and 27.1), with one woman considered obese (BMI = 32.7).

Of the 15 women, mothers of infants, no one suffered from undernutrition. In contrast two women from *Poxo Muybu* and two from *Sawré Muybu* presented as overweight (BMI ≥ 25.0). The BMI median was 23.6 kg/m^2^ (IQR = 20.53, 23.92, and 25.91), with six (38%) found to be overweight (Table 1).

Moreover, the analyses showed that for three of the four stunted children, the mother was found to be overweight (BMI 26.6, 26.3 and 29.2).

A Kruskal–Wallis test did not suggest any significant differences in Infant Hb levels between villages [H (2) = 2.7, *p* = 0.255]. An analysis of the relationship between Mother’s H-Hg level and infant’s Hb level did not find any significant correlation (Spearman’s r = 0.073, *p* = 0.831), nor was any significant correlation seen between Infant H-Hg and Hb levels (Spearman’s r = −0.265, *p* = 0.431). In our small sample no relationship was identified between Infant growth and Infant Hb (Spearman’s r = −0.146, *p* = 0.668).

### 3.4. Hair Methylmercury Levels in Women of Childbearing Age

Comparing hair methylmercury (H-Hg) in all women of childbearing age to the reference limit of 6.0 µg/g H-Hg, women’s levels were significantly greater (median = 7.115, IQR = 4.678, Wilcoxon’s rank = 416, *p* < 0.008). This group had an average of 7.71 µg/g H-Hg with a range 2.00 to 20.19 µg/g. The lowest H-Hg level (2.44 µg/g) was a 24-year-old woman from *Sawré Muybu (SM)* who described eating very little meat and fish, whilst the highest level (13.8 µg/g) recorded in a 19-year-old woman from *Sawré Aboy (SA)* was noted to consume fish frequently (≥3 times/week).

Amongst the 15 mothers who had given birth within the last two years, nine (60%) were found to have H-Hg levels above 6.0 µg/g, with an average of 7.56 µg/g, range 2.44 to 13.81.

In the worst affected region, *Sawré Aboy* (SA), the average H-Hg level in women of childbearing age was 12.67 µg/g, twice our Amazonian reference level of 6.0 µg/g. Average H-Hg amongst women of childbearing age in *Poxo Muybu* (PM) and *Sawré Muybu* (SM) also exceeded our 6.0 µg/g reference limit, at 7.57 µg/g and 6.32 µg/g, respectively (See Figure 2).

The five women pregnant at the time of analyses ranged in age from 15 to 39, all from the *Poxo Muybu* community. All presented with H-Hg levels above 6.0 µg/g reference level. The highest H-Hg recorded in this subsample was 12.9 µg/g in a 24-year-old woman, 12 weeks’ gestation and parity two.

### 3.5. Hair Methylmercury Levels in Infants

Examining all infant H-Hg levels we found median levels were 3.88 (IQR = 3.05 µg/g, Wilcoxon’s rank = 5, *p* = 0.0006). Three of the 16 infants had H-Hg levels over 6.0 µg/g. The highest level (19.6 µg/g) of mercury was found in an infant aged 11 months in the village of *Sawré Muybu* (*SM*) (See Table 1).

On analysis of Mother’s H-Hg concentration in relation to their infant’s H-Hg, we found no significant relationship between those infants solely breastfed in their first six months of life. When exploring the whole cohort those mothers with an H-Hg above 6.0 µg/g showed a positive association with their infant’s H-Hg (*r* = 0.783, *p* (2-tailed) = 0.01).

### 3.6. Neurodevelopment Results of the Infants Studied under Two

Three of the 15 infants failed the Denver II test with two children failing on language and one on gross motor skill. These children’s H-Hg levels were 2.59 µg/g, 2.44 µg/g and 19.58 µg/g with the relevant maternal H-Hg levels recording 7.25 µg/g, 4.31 µg/g and 6.18 µg/g respectively.

### 3.7. Comparison between Villages

Analysis with Kruskal–Wallis showed that population levels of H-Hg differed significantly between the villages, both between women of childbearing age and between infants under 2 years old [H(2) = 15.0, *p* = 0.0005 and H(2) = 6.2, *p* = 0.0455], respectively.

Post-hoc analyses of women of childbearing age suggests that *Sawré Aboy* had significantly higher H-Hg levels (median = 12.177, IQR = 3.017), than either *Poxo Muybu* (median = 7.115, IQR = 2.832; Mann-Whitney’s U = 32.0, *p* = 0.0041), or *Sawré Muybu* (median = 4.508, IQR = 5.552; Mann-Whitney’s U = 32.0, *p* = 0.0009). Between *Poxo Muybu* and *Sawré Muybu* villages there was no statistical differences (Mann-Whitney’s U = 141.0, *p* = 0.0344).

Post-hoc analyses of infants under 2 years old revealed that levels of MeHg in *Sawré Aboy* (*n* = 5, median = 5.355, IQR = 1.284 µg/g) were significantly higher than *Poxo Muybu* (*n* = 5, median = 2.345, IQR = 0.720 µg/g; Mann-Whitney’s U = 2.0, *p* = 0.0367), and *Sawré Muybu* (*n* = 5, median = 3.977, IQR = 2.503 µg/g; U = 10.0, *p* = 0.6761), whilst a trend suggests higher levels are also present in *Sawré Muybu* than *Poxo Muybu* (U = 22.0, *p* = 0.0601). Figure 2 shows the distribution of measurements of MeHg among the three villages that were included in the study.

## 4. Discussion

In response to the specific request from community leaders, this study brought together a health evaluation and an assessment of methylmercury exposure for women of childbearing age and their infants under two years old in a particular Munduruku Indigenous Land never before studied.

Our findings demonstrated that not only women of childbearing age but also their infants have been facing colossal public health challenges. These challenges are consequences of increasing illegal mining activities in their territory and due to historical abandonment by the Federal Government. Among the principal issues illustrated in this study, we highlight high levels of methylmercury exposure, stunting, anemia, poverty, and food insecurity, as well as low vaccination coverage and high infant mortality rates.

### 4.1. H-Hg in Women of Childbearing Age

Our study identifies a population chronically exposed to unsafe levels of MeHg, with the average hair MeHg concentration sitting above our reference limit of 6.0 µg/g. It is clear that a severe public health concern exists regarding the future generations of Munduruku, with the population of *Sawré Aboy* at greatest risk, recording an average hair MeHg level in women of 12.1 µg/g. Levels amongst these women mirror the levels found in other communities exposed to mercury released by ASGM in the Tapajos River basin [33,66,67,68] as well in the broader Brazilian Amazon region [69,70,71,72].

A wide range of mercury levels have been identified in different populations across the world, associated with a high fish diet. For example, Caribbean immigrants living in Brooklyn-NY [73] showed mercury levels in the blood of 2.14 µg/L, which is equivalent to 0.43 µg/g hair Hg (according to the WHO`s conversion measure [61]). Bjornberg et al. [74], studying pregnant women from Sweden, found average mercury levels in hair equal to 0.35 µg/g. These groups have hair methylmercury concentrations below the limit proposed by U.S.EPA (i.e., 1.0 µg/g in hair samples which derives of the Reference Dose of 0.1 µg MeHg/kg bw/day) [45]. On the other hand, the indigenous communities from Canada investigated by Muckle et al. [75] showed hair mercury levels of 4.4 µg/g. Furthermore, a study conducted in New York by Fletcher and Gelberg [76] involving a wealthy urban population revealed mercury levels in blood of 30.8 µg/L, corresponding to 6.16 µg/g in hair. This result contrasts with our findings, where mercury levels above the reference level (i.e., 6.0 µg/g) in the Munduruku indigenous peoples of the Middle Tapajós can be explained by limited access to other animal protein sources. This dietary limitation is not a likely explanation for the mercury levels observed in investigated New Yorkers whose income is relatively high. It is possible that the high levels of mercury observed in the urban population of New York (i.e., above 6.0 µg/g) are the result of polymorphisms in genes that contribute to the toxicokinetics and toxicodynamics of mercury [77,78,79]. Perini et al. [80] investigated genetic polymorphisms in Munduruku indigenous people and identified two individuals with alterations in genes (*ALAD*) that hinder the elimination of mercury by the organism, who also present high levels of mercury in their hair.

The Munduruku communities represented in the present study have an elevated fish consumption as can be accessed in the recently published paper by Vasconcellos et al. [12]. This investigation concluded that women of childbearing age consume around 170 g of fish daily and consequently have a MeHg intake dose of 0.73 µg/kg bw/day (in the current exposure scenario built). This MeHg intake dose corresponds to seven times higher than the reference dose proposed by U.S.EPA [45] and three times higher than the safe dose established by FAO/WHO [46].

In line with our findings, a Metareview across 72 countries found that hair methylmercury levels largely fall below 2 µg/g and identified communities living alongside ASGM as at-risk populations for high mercury exposure [81]. Chronic MeHg exposure in adults, even at lower levels than those reported in the current study, has been associated with attention deficits and interruptions in fine-motor function as well as sleep disturbances, fatigue and depression with the impact on the aging brain the subject of recent concern [82,83,84].

### 4.2. H-Hg in Infants

Median H-Hg levels (3.80 µg/g) found amongst Munduruku infants reflect those found in other regions of the Amazon [41,43,85,86,87]. Santos Lima et al. [41] reported an association between MeHg exposure and lower performance in neuropsychological tests for children and adolescents, with each 10 μg/g increase of hair Hg corresponding to poorer performance by half a standard deviation. In this study, three infants failed the Denver II Developmental Screening Test, two on Language and one on Gross-Motor Skills, all these infants have high levels of H-Hg. Low-level pre-natal exposure to MeHg through maternal fish consumption has been associated with poor performance in Denver II tests [88,89] whilst other studies have found no association [90]. Due to a small sample, it is difficult to draw conclusions from our findings, collected by one pediatrician in Portuguese and not Munduruku native tongue. However, Gross Motor deficits are less subject to bias. Thus, while infant H-Hg levels are ostensibly lower than those found amongst adult women, it is important to remember the vulnerability of infants to environmental threats, given their immature immune systems and continuous growth [91,92].

### 4.3. Interactions between Maternal and Infant H-Hg

Studies have shown a significant relationship between maternal methylmercury levels and infant methylmercury levels from birth and during the first years of life due the intrauterine transfer and the breastfeeding [38,39,40,61]. Although the concentrations of mercury in breast milk are low, breastfeeding promotes continued exposure to mercury during the first months of life [38,39,40,61]. The intrauterine exposure is deemed a far greater risk to the developing fetus, particularly during the first trimester, since the MeHg levels in the infant blood are almost twice that of the levels found in their mothers’ blood [93]. After birth, the infant MeHg levels begin to fall, suggesting that the exposure from breastmilk is lower [85,93,94,95,96]. Our study did not show a significant correlation between the methylmercury detected in mothers and their infants aged under six months who were solely breastfed. This may be in part to the limited power of our cohort to establish robust associations, but could also be explained by the reduced methylmercury exposure from the breastfeeding. This may have allowed infant methylmercury levels to fall following birth [94]. In the context of this indigenous Munduruku community, this finding could be interpreted as a positive message for the Munduruku women, that they can be less concerned about infant mercury exposure during the initial months of breastfeeding, where the health benefits of the maternal milk (including neurodevelopmental benefits) are higher compared to the risks posed by mercury during breastfeeding [85].

The statistical analyses did show a significant association between maternal and infant H-Hg levels amongst mothers with H-Hg levels above 6.0 µg/g. This association may suggest an increase in progressive methylmercury exposure for the infants the higher the mother’s MeHg burden. However, additionally, the H-Hg infant levels could relate to the gradual integration of the developing infant into household eating habits [85]. So, the higher H-Hg levels in mothers represent a household more reliant on subsistence fish consumption [85].

### 4.4. H-Hg by Village

The average methylmercury levels detected in the studied participants varied significantly among the villages investigated. The *Sawré Aboy* village showed significantly higher levels of MeHg than the other two communities, exceeding even the highest Benchmark Dose for pregnant women found in current literature (i.e., 12 µg/g) [97]. With these higher levels, the health implications for this community could be particularly serious.

*Sawré Aboy* village is found near to the meeting of the Jamanxim and Tapajos Rivers. This village is the most isolated of the three villages and is furthest from any access to the largest town, Itaituba (Pará State). The habitants of this village catch the majority of fish for consumption in the Jamanxin river, which is severely impacted by the ASGM. The distance to major towns and the intense gold mining activity in this locality reduce the access to a varied diet and, at the same time, increase the exposure to mercury released by the mining [98]. These factors combined may well explain the elevated mercury detected in the indigenous living in *Sawré Aboy* village. Other studies that looked at nearby communities are consistent with our findings, showing high levels of methylmercury in the community and, in addition, a significant correlation with neurotoxicity [99].

### 4.5. Nutrition and Growth

The prevalence of infant anemia amongst these three Munduruku communities investigated is 50%. This value is considered a severe public health crisis by the WHO [100] and should be the target of intervention. The infants under two are known to be a high-risk group given the high demands for iron, folate and Vitamin B12 during this period of growth as well as an increased frequency of infections and parasitic disease [17]. In indigenous Brazilian children, the proportion of anemia is commonly found to be around 50% [17]. This prevalence is twice that found in comparable non-Indigenous populations [17]. This disparity is observed in indigenous communities worldwide and is often associated with preventable risk factors such as food insecurity, poor living conditions and sanitation, as well as the higher prevalence of malaria and intestinal parasites [101]. Anemia is known to negatively impact cognitive development, physical growth and immunity and is likely to further aggravate the vulnerability of this population to poverty and illness [102].

Infants identified with a slower weaning regime from six months (i.e., weekly as opposed to daily alternative foods) were found to have a higher prevalence of anemia. However, it is important to bear in mind that breastfeeding is a protective factor against neurodevelopmental delays and increases the odds of infant survival [85,103]. An exclusively breastfed child is 14 times less likely to die in the first six months of life than a non-breastfed child [104]. However, it is understood that prolonged breastfeeding without sufficient iron supplementation leaves the developing infant at risk of Iron Deficient Anemia, and it would be important to know whether the families of slow weaned infants have access to sufficient alternative foods to allow a secure and consolidated weaning regime or importantly whether we are identifying gaps in maternal awareness and understanding of this topic [86,105,106,107].

The prevalence of anemia in mothers and women of childbearing age sits at 24% and 15% respectively, making them of moderate and mild public health significance. Anemia during pregnancy is associated with poor pregnancy outcomes such as preterm birth and low birth weight as well as maternal and fetal mortality, and the high iron demands of pregnancy are likely to aggravate preexisting anemia [108,109,110,111]. WHO estimates of anemia amongst women of childbearing age in Brazil 2009 were 16.1%, suggesting that the findings in these Munduruku communities mirror the situation at a national level [112].

Establishing anemia through Hb levels alone makes it impossible to isolate its exact cause. These communities have limited access to a varied diet, making food insecurity and micronutrient (e.g., iron, folate) deficiencies more likely [16]. Poor sanitation and access to healthcare, alongside a high incidence of intermittent malaria, intestinal parasites and diarrhea caused by gastrointestinal infections, pose further risk factors to the development or aggravation of anemia [113,114].

The women of childbearing age from the communities investigated showed high prevalence of overweight (38%). Studies of indigenous communities in Brazil have found a wide range in the prevalence of overweight [115,116]. The First National Survey of Indigenous People’s Health and Nutrition in Brazil finding the overweight prevalence of 30% [16]. The comparisons are hampered by the varied state of isolation, with some communities remaining isolated and others integrated into the local area. The Munduruku communities studied here remain semi-isolated with daily life spent within the communities and as such levels of physical inactivity and genetic predisposition should be explored [16].

On the other hand, infant Z scores showed several infants with restricted growth. Height for age (HAZ) scores showed 22% infants were moderately-severely stunted, closely mirroring the findings of the First National Survey of Indigenous People’s Health and Nutrition in Brazil [16]. This is in comparison to 2.2% expected in a well-fed population. Stunting in indigenous communities has been found to reflect a combination of serious nutrient deficiencies and chronic infection such as parasitic infections, which pull on nutritional reserves and are rampant in environments where good sanitation is a challenge [117]. The prevalence of stunting within this small cohort therefore identifies serious nutritional deficits, leading to gaps in brain development and future successes in school, as well as earning capacity in adult life. In addition, there are implications for those non-stunted developing infants within a community that presents with stunting, where these infants are likely affected by under-nutrition [118]. Furthermore, for women growing up stunted there are serious consequences during childbirth, with a higher likelihood of obstetric complications, largely due to restricted pelvic growth [118].

It is worth noting that of the four infants with stunted growth, three have over-weight mothers, with the infant of the most overweight mother also considered under-weight. The so called ‘double-burden of disease’ where overweight and undernutrition are found within the same household, is a common finding amongst Indigenous communities experiencing food insecurity and undergoing a nutritional transition: moving away from a traditional Indigenous diet and increasingly eating a high-fat westernized diet [117,118,119,120,121,122]. This finding is surprising in a hard-to-reach region, where communities are ostensibly still reliant on Indigenous grown foods. However, this change in dietary pattern has been noted in Amazonian riverine communities (also called *ribeirinhos* in Portuguese) and as a result of the ever-increasing presence of gold miners coming from across Brazil, the rapidly increasing rates of deforestation and the Trans-Amazonian Highway making access easier from the outside, the nutritional transition seen across Brazil may be expanding [10,123,124]. It is worth noting that the pattern of overweight and undernutrition was not found for the stunted infants in *Sawré Aboy*, the most isolated of the communities, but was identified in *Poxo Muybu* and *Sawré Muybu* which have access to Itaituba and the Trans-Amazonian Highway, respectively.

It is clear that the nutritional situation in these communities is quite poor, with profound, lifelong, and irreversible consequences for the developing Munduruku infant. The implications for the developing brain of a 50% prevalence of anemia in these infants are catastrophic and likely irreversible [125,126]. Good nutrition in the first 1000 days of life is recognized globally as a key intervention to help a child’s ability to grow, learn, and rise out of poverty [118,127]. An intervention is needed to support this population’s long-term health, stability, and prosperity.

### 4.6. Infant Mortality

The Infant Mortality Rate (IMR) of 86.7/1000 live births estimated in this study for the three Munduruku villages is higher than the regional average for indigenous communities of 50.62/1000 lives births recorded in 2012, but is similar to the Kaiapó indigenous community (neighbor to the studied villages), which recorded an IMR of 85.3/1000 in 2018 [128,129]. Indeed, the research on infant mortality among indigenous children conducted by Lima et al. [128] and Teixeira et al. [129] found the majority of reported infant deaths amongst indigenous groups to be from Kaiapó and Munduruku ethnicities. This scenario suggests an ongoing burden on resources for maternal health within these communities [128,129]. With Brazil’s national IMR at 12/1000 live births it has become well established that indigenous communities in South America present with a higher infant mortality rate than the national average [130].

It is important to note that the higher rates recorded within our study community may in part reflect the data collection, since our study reflects personal information gathered in fieldwork interviews. We do note that we do not have annual data to define a specific yearly incidence.

We cannot specifically comment on the causes of the infant deaths within this community, but the main causes of infant deaths in the Pará state have been identified as difficulties during birth, respiratory tract diseases and infectious and parasitic diseases [131]. All the causes are largely preventable with access to simple, affordable interventions including immunization, adequate nutrition, safe water and food and quality care by a trained health provider when needed [131]. The evidence has highlighted the difficulties in making timely interventions during the perinatal period for indigenous populations with poor access to health care [132,133]. In addition, the way of life in a village with reduced access to clean water, poor sanitation and minimal health care resources allows parasites, infectious diseases and malnutrition to persist, putting infants within our studied community at risk of increased mortality post-partum [134,135]. Furthermore, studies following the Minamata tragedy in Japan suggested that rates of miscarriage and stillbirths increased following exposure to mercury [134,136]. Benefice et al. [137] found that women living along the Beni River in Bolivia with MeHg levels over 5 µg/g presented with increased rates of infant deaths but established no causality link between MeHg exposure and infant deaths.

Our study identifies gaps in resources and infrastructure for reducing these often-reversible causes of infant death. Brazil has made good progress in reducing infant mortality but there is not the same pattern within indigenous communities only confirming major ongoing deficits in health care provision within these communities of difficulty access [22].

Another critical problem detected in our study was the insufficient vaccination coverage for this specific age group. All infants enrolled were missing vaccines. It is worth remembering that vaccination opportunity and coverage are fundamental issues for public health, especially during pandemics. Although the literature is scarce on vaccination coverage among indigenous peoples in Brazil, the consequences of low vaccination coverage on the infant mortality rates due to vaccine-preventable diseases are widely known [138,139]. At least part of the health challenges observed in the studied population may result from poor vaccination.

Within a resource depleted system, even thorough community-based intervention programs can struggle to reduce the IMR burden [140,141]. The high rates of infant mortality observed in this study are a consequence of the restricted and reduced access to secondary healthcare services, poor sanitation, and malnutrition. This study highlights the importance of thorough pre-natal care to identify women at risk of birth complications, increased training of local health care professionals in management of peri-partum care and a focus on nutritional supplementation for mothers and infants to avoid poor development and weak immune systems. Further guidance on sanitation can help reduce risks to infant mortality in their first year of life, from diarrhea and parasitic infections through clean water supplies, closed toilet systems, education regarding hygiene and wider access to anthelminthic medications.

### 4.7. Where Can We Go from Here?

*“*Women and their families should receive support to improve their diets as a general health rule, which is a basic human right” [142].

We encourage the Brazilian government to provide adequate investment to the National Unified health system (SUS) and the Special Indigenous Health Secretariat (SESAI) to screen these populations for basic needs and identify those households at greater risk of poor health outcomes. The Brazilian government committed to five pillars to improve the nutritional status of the country alongside the WHO by 2019, including commitments to focus on rural families, but these indigenous communities may be getting left behind [143].

The geography of the land and the isolated nature of living will always make service provision harder; however, evidence-based strategies can work to help improve health indicators in rural communities [141].

Finally, the Covid-19 pandemic has encouraged innovation in providing access to health through ‘Telemedicine’, a resource that could overcome some of the barriers to improving the social determinants of health in these communities [144].

The WHO and UNEP has described measures to communicate risk to vulnerable populations and produce an effective mercury intervention program:

“The risk manager needs to characterize the role of fish in the population of concern. Fish may play a large role in the cultural and socio-economic fabric of the country or region. The risk manager needs to evaluate whether there are alternative foods that are readily available, affordable and of equal nutritional benefit. There may be other risks associated with alternative fish or foods that should be identified and evaluated” [61].

With the above guidance in mind, we propose several evidence-based recommendations to help improve this chronic and severe public health crisis both in relation to mercury and, more broadly, nutrition. It is essential that we implement these recommendations, paying close attention to the cultural characteristics and respecting the traditional values of the investigated communities.
Encourage training and education in nutritional best practices with a focus on the 1000 days window;Improve education around breastfeeding and encourage comprehensive weaning practices post six months;Identify those infants at risk of anemia, for example those with extended breastfeeding, and provide sufficient nutritional drinks and iron supplementation;Work closely with the National Unified health system and the Indigenous Special Sanitary Districts to help develop continuous, accessible, and culturally sensitive pre- (greater than four visits) and post-natal health care;Help develop methods to specifically catch those fish high in Omega-3′s and low in MeHg (i.e., those lower in the food chain);Encourage the Brazilian government to align with the Minamata convention and Planet Gold’s global initiative to reduce Mercury pollution in ASGM sites;Condemn the government for its active encouragement of illegal gold mining in the Munduruku area.

### 4.8. Limitations

An important limitation to note is our small sample size of mothers and their infants. This reduces our power to detect any associations; however, it reinforces the need for further, more comprehensive studies in the future.

We acknowledge that in environments where one cannot control for the numerous covariates that are known to adversely influence child neurodevelopment it is not possible to accurately determine causal relationships and, therefore, we further clarify that we present only associations.

The present study is based on evidence from previous literature that mercury is passed from mother to child in utero and, although to a lesser degree, also during breastfeeding. We are limited by our dataset to analysis of the transmission from mother to child during pregnancy and at birth. While we did not have data on cord blood mercury levels, research does show hair sample levels are of correlating value.

In addition, we did not have access to sufficient data on infant birth weight, which could have helped with further analyses regarding prenatal health indicators, nutritional status and infant growth. It would also have been useful to include a control group population, however, the possibility of this has been noted in previous literature to be difficult [10].

Finally, we refer to the number of infant deaths described by the women within this population, however we do not have further details in order to establish annual changes in this figure. We also lack data regarding the mechanism of death; however, our figures are not dissimilar to the figures reported for indigenous communities within the region and rates remain consistent within age cohorts, nevertheless it is clear further investigation is warranted.

## 5. Conclusions

This study was borne out of a request from the leaders of these Munduruku communities, exemplifying the experience of vulnerability within the communities;These already vulnerable Munduruku communities suffer chronic exposure to levels of MeHg above 6.0 µg/g reference levels, known to negatively impact on adult health and infant growth and development;Ongoing health inequalities are contributing to higher rates of anemia, stunting and infant mortality, preventing individuals in these Munduruku communities from reaching their full potential;Simple, culturally sensitive intervention is needed to reduce poor health outcomes and interrupt this cycle of poverty.

## Figures and Tables

**Figure 1 ijerph-18-10091-f001:**
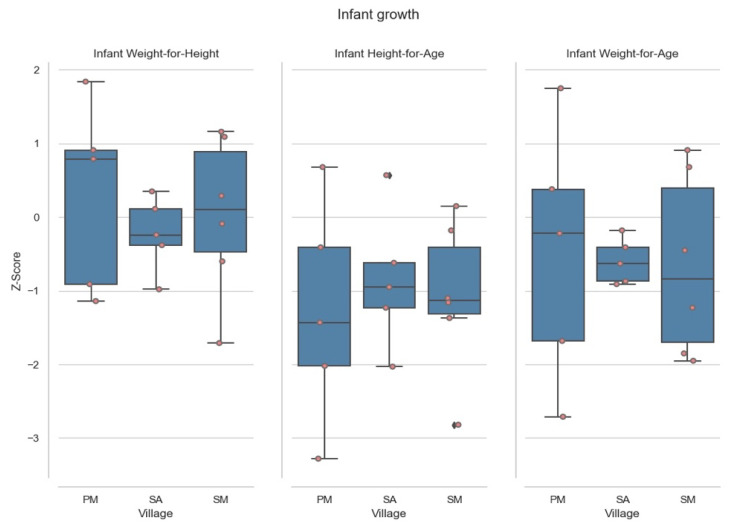
Infant Z Scores for measures of growth. PM corresponds to *Poxo Muybu* village; SA is *Sawré Aboy* and SM is *Sawré Muybu*. S.D. < −3 is severely deficient (severely wasted/stunted/underweight); S.D. < −3 and < −2 correspond to deficient (wasted/stunted/underweight); zero is normal; >2 is weight-for-height is overweight, *Sawré Muybu* Indigenous Land, Brazilian Amazon, 2019.

**Figure 2 ijerph-18-10091-f002:**
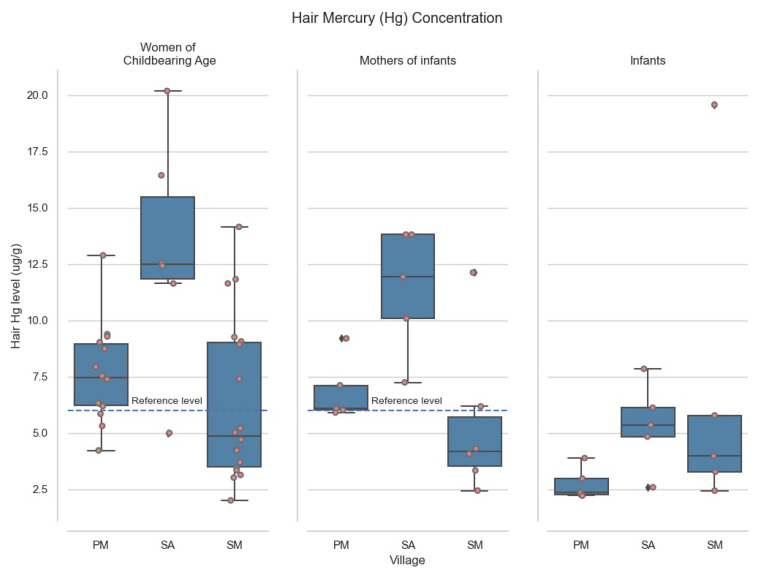
Methylmercury levels split by villages. PM correspond to *Poxo Muybu* village; SA is *Sawré Aboy* and SM is *Sawré Muybu*. *Sawré Muybu* Indigenous Land, Brazilian Amazon, 2019.

**Table 1 ijerph-18-10091-t001:** Demographic (sex and age) and clinical (methylmercury and hemoglobin levels, as well nutritional status) characteristics of mothers and their children under 2 years, *Sawré Muybu* Indigenous Land, Brazilian Amazon, 2019.

Mothers to Infants ID	H-Hg (µg/g)	Age (Years)	Mother’s Hb (g/dL)	Mother’s BMI (kg/m^2^)	Infant ID	Sex	Age (months)	H-Hg (µg/g)	Hb (g/dL)	Height for Age (Z-Score)	Weight for Age (Z-Score)	Denver II Passed (Y/N)?
*Poxo Muybu*												
PM-01-03-01	6.02	16	12.1	19.6	PM-01-03-01-01	M	9	2.25	12.2	−1.43	−0.22	Y
PM-03-01-02	6.08	45	11.9	26.3	PM-03-01-02-05	F	20	2.97	12.1	−2.02	−1.68	Y
PM-05-03-01	5.91	22	14.6	24.0	PM-05-03-01-02	M	8	2.35	10.3	0.68	1.75	Y
PM-06-01-02	7.12	31	12.5	29.2	PM-06-01-02-03	M	21	2.22	11.1	−3.28	−2.71	Y
PM-13-02-02	9.20	19	12.4	19.0	PM-13-02-02-02	F	5	3.88	-	−0.41	0.38	Y
*Sawré Aboy*												
SA-01-01-02	10.1	19	11.4	19.6	SA-01-01-02-02	F	5	5.36	-	−0.62	−0.63	Y
SA-04-01-02	12.0	17	12.6	19.9	SA-04-01-02-01	F	19	7.85	11.3	−2.03	−0.87	Y
SA-06-01-02	13.8	19	14.4	24.4	SA-06-01-02-01	M	21	6.12	11.3	−1.23	−0.91	Y
					SA-06-01-02-02	F	5	4.84	-	0.57	−0.41	Y
SA-07-02-02	7.25	19	11.9	20.4	SA-07-02-02-02	F	21	2.59	10.7	−0.95	−0.18	N
*Sawré Muybu*												
SM-01-02-02	12.1	23	13.8	21.4	SM-01-02-02-04	M	9	3.28	10.4	−1.10	−1.85	Y
SM-04-03-02	4.10	18	13.4	26.3	SM-04-03-02-03	F	7	-	9.80	−2.81	−1.95	Y
SM-12-01-02	6.18	16	11.1	25.6	SM-12-01-02-01	F	11	19.6	10.7	−1.37	−0.45	N
SM-14-01-02	2.44	24	13.4	25.8	SM-14-01-02-04	M	7	5.78	9.40	−0.18	−0.18	Y
SM-15-01-02	4.31	16	12.8	24.2	SM-15-01-02-02	F	9	2.44	12.0	0.15	0.91	N
SM-20-01-02	3.34	29	15.3	23.6	SM-20-01-02-05	F	5	3.98		−1.16	−1.23	Y

**Table 2 ijerph-18-10091-t002:** Descriptive analysis of the woman of childbearing age, according to the methylmercury exposure levels (<6.0 μg/g or ≥6.0 μg/g), *Sawré Muybu* Indigenous Land, Brazilian Amazon, 2019.

	Methylmercury Levels	
	<6.0 μg/g	≥6.0 μg/g
[MeHg]-Mean (μg/g)	4.1 (*n* = 20)	9.9 (*n* = 33)
Age Group (years)		
12–18 (*n*)	9	12
19–30 (*n*)	9	13
30–49 (*n*)	2	8
**Sociodemographic Characteristics**		
Salary Mean (US$)	281	217
Education Level (median schooling years)	6	6
Household (*n*)	11	8
Agriculture (*n*)	3	11
Student, Teacher (*n*)	5	6
Partner working as extractivist (*n*)	0	5
**Dietary Characteristics**		
Frequently Fruit Intake (*n*)	20	33
Weekly fish consumption (Median)	3	3
Nuts consumed in the wet season (*n*):		
*Daily*	10	19
*Weekly*	7	5
*Monthly*	1	8
**Health Outcomes and Obstetrician History**		
Hb Median (g/dL)	13.4	13.2
BMI (kg/m^2^)	23.2	22.8
Depressed mood (*n*)	8	8
Live Births (*n*)	53	97
Miscarriages (*n*)	1	5
Infant deaths (*n*)	5	8

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
