# Peer review of "An Assessment of Health Outcomes and Methylmercury Exposure in Munduruku Indigenous Women of Childbearing Age and Their Children under 2 Years Old"

_ijerph, 2021, doi:10.3390/ijerph181910091_

Round 1
Reviewer 1 Report
In this paper, hair and haemoglobin samples and related data were collected from women of child bearing age and infants under 2 years of age in three semi-isolated Munduruku communities along the Tapajos River. The results were statistically analyzed to obtain the local distribution characteristics of MeHg and the relationship between MeHg levels and infant growth in women of childbearing age. It is clear that the authors’ workload is large and detailed, but the paper is not innovative and the analysis of the data is not deep enough. At the same time, the data is only analyzed for its relevance and does not provide suggestions to the local residents on how to reduce health risks, which is not practical. In this regard, I have some comments and suggestions for the authors to make relevant corrections to the article
- None of the figures in this article are clear enough, please make adjustments;Lines 249-250,this sentence has been split into two parts by Figure 2, please adjust the position of Figure 2.
- The Mothers to Infants ID and Infant ID in Table 1 are not in the same row and the units kg/m2 have a formatting error. Please adjust the text in the table to make it more aesthetically.
- Introduction lacks background information, please indicate what level of contamination the MeHg concentrations selected for this paper are in other regions.
- In section 2.1, three communities are selected for this paper, but only the importance of Sawré Muybu is explained. Are these the only three communities in the area, or please indicate the representativeness of the three communities selected.
- Lines 91-92, ‘Childbearing age’ is 12-49, but as far as I know it is 15-49, is childbearing age here documented or is it the custom of the local residents?
- Lines 294-300 point out the controversy over safe exposure levels in utero, but it is not clear which side is supported by the results of this article.
- The sentence in lines 299-300 mentions the effects of fish consumption, but this sentence is somewhat abrupt here and is suggested to put it in the relevant section of 4.2.
- Please analyze the results of the article in more depth and give practical suggestions for the local residents to avoid the dangers of MeHg. Relevant suggestions can be made in terms of local government measures and the diet adjustment plan of the residents.
- Please provide a summary of the innovation of this paper, preferably comparing it with previous research to show the superiority of the article.
Author Response
Dear Editor in chief and Dear Reviewer 1,
We hope this message finds you well.
Concerning the review report form about the manuscript “Chronic methylmercury exposure in Munduruku indigenous women of childbearing age and its consequences for their children under 2 years old - ijerph-1283940”, we would like to thank you for the valuable technical support and the questions pointed by the reviewers.
Following your suggestions, we promote a comprehensive revision in the paper, including changes in title, general objective, results, and discussion. Please see below, the point-to-point response to the reviewer’s comments.
We try to accommodate all reviewers' questions in order to reach an innovative approach and perform a deeper analysis. Moreover, at the end of the paper, we included a piece of suggestions to the residents on how to reduce health risks.
We hope that our manuscript has enhanced quality and is now more suitable to publish in the International Journal of Environmental Research and Public Health (IJERPH).
Please, let us know if you have additional questions.
Kind regards
Comments and Suggestions for Authors
In this paper, hair and haemoglobin samples and related data were collected from women of child bearing age and infants under 2 years of age in three semi-isolated Munduruku communities along the Tapajos River. The results were statistically analyzed to obtain the local distribution characteristics of MeHg and the relationship between MeHg levels and infant growth in women of childbearing age. It is clear that the authors’ workload is large and detailed, but the paper is not innovative and the analysis of the data is not deep enough. At the same time, the data is only analyzed for its relevance and does not provide suggestions to the local residents on how to reduce health risks, which is not practical. In this regard, I have some comments and suggestions for the authors to make relevant corrections to the article
- None of the figures in this article are clear enough, please make adjustments;Lines 249-250,this sentence has been split into two parts by Figure 2, please adjust the position of Figure 2.
Reply: Thank you for the comment. Please see new formatting of figures. We hope they are clearer.
- The Mothers to Infants ID and Infant ID in Table 1 are not in the same row and the units kg/m2have a formatting error. Please adjust the text in the table to make it more aesthetically.
Reply: This table has now been adjusted to ameliorate any errors.
- Introduction lacks background information, please indicate what level of contamination the MeHg concentrations selected for this paper are in other regions.
Reply: The introduction has been improved to provide clearer background information, also included is a further in-depth section at the start of the discussion looking at how these concentrations compare locally and more broadly around the World.
- In section 2.1, three communities are selected for this paper, but only the importance of Sawré Muybu is explained. Are these the only three communities in the area, or please indicate the representativeness of the three communities selected.
Reply: The Sawre Muybu indigenous land is a large area of land which includes 8 villages. The Sawre Muybu village included in the study is one, eponymously named village, the other two villages we studied (Sawre Aboy and Poxo Muyubu) are also within the Sawre Muybu indigenous lands. They are representative of the other 5 communities and in total the region includes around 800 Munduruku. This is now detailed in the paper.
- Lines 91-92, ‘Childbearing age’ is 12-49, but as far as I know it is 15-49, is childbearing age here documented or is it the custom of the local residents?
Reply: It is custom in indigenous communities in the Amazon to have an early age of sexual initiation, it is not uncommon to see families starting with mothers as young as 12, many are already married. Two previous papers regarding indigenous women of childbearing age have chosen an initial age of 10 and 14. We decided to choose a middle point. We have explained this in the methods with the mentioned paper referenced. (Estima et al 2018 and Garnelo L et al. 2019)
- Lines 294-300 point out the controversy over safe exposure levels in utero, but it is not clear which side is supported by the results of this article.
Reply: Thank you for the comment. There does exist a continuous debate to strike the right balance between the benefits of fish consumption in pregnancy and the risks of mercury exposure. We hope this now appears clearer.
- The sentence in lines 299-300 mentions the effects of fish consumption, but this sentence is somewhat abrupt here and is suggested to put it in the relevant section of 4.2.
Reply: As mentioned before, we promote a comprehensive revision in the paper, including changes in title, general objective, results, and discussion. We hope we have cleared up the discussion.
- Please analyze the results of the article in more depth and give practical suggestions for the local residents to avoid the dangers of MeHg. Relevant suggestions can be made in terms of local government measures and the diet adjustment plan of the residents.
Reply: As mentioned before, we try to accommodate all reviewers' questions in order to reach an innovative approach and perform a deeper analysis. Please see the end of the discussion in 4.7, we have referred to guidance suggested by the WHO, as well specifically outlining recommendations.
- Please provide a summary of the innovation of this paper, preferably comparing it with previous research to show the superiority of the article.
Reply: Firstly, we have adapted our paper to create a study presenting broader health indicators which we feel now provides a more novel, more informative, and useful contribution for the Munduruku population, relevant health services and the wider scientific community. Feedback from the second reviewer regarding methylmercury (MeHg) exposure and infant growth suggested the data was not robust. We agree the power of this study was small and therefore it has been removed from the paper, further investigation in future studies will explore these initial findings further with more participants. For now, we have focused on presenting a comprehensive analysis of the health status of this community using MeHg exposure, malnutrition and maternal health indicators. Our findings show clear vulnerabilities and urgent action.
There is one other paper with a similar method published in 2013 (Coimbra et al. 2013, referenced in our paper), however given the isolation, variety of cultural practices and heterogeneity in healthcare provision and infrastructure of indigenous communities in all Brazilian territory, studies of this nature are far more relevant, informative and much needed on a smaller scale. The innovation of our paper lies in the intimate focus of one indigenous community regarding risks to development, morbidity and mortality for future generations of Munduruku through MeHg exposure and undernutrition alongside deficits in health and sanitation services.
We present a study on MeHg exposure in the Brazilian Amazon in collaboration for the first time with British or even European universities. Santos-Sacremento et al. (2021) state specifically that international collaboration would enhance dialogue and recognition of this subject within the international scientific community.
This study was requested by local leaders and supported by the Local department of indigenous health showing the heightened vulnerable state they are in and the necessity of this study. This is alongside the very real current political vulnerability indigenous communities are facing (Crespo Lopez et al. 2021). This community has never been analyzed before and it is important to continue to map the burden of mercury exposure within this region (Santos-Sacremento et al. 2021). Further to this, we are showing particularly high levels of hair mercury levels which are above global BMDL’s (Benchmark Dose Levels) (NRC 2001), posing a significant risk to future generations of Munduruku.
Further innovation of this paper lies in the thorough assessment of health indicators of all infants and women of childbearing age (census) living in the Sawre Muybu Indigenous Land, with deep analyses of the developing infant’s nutritional status and of maternal health.
The novelty lies in the wide scope of this paper presenting a significant burden of anemia in infants, a high prevalence of stunting and low vaccine coverage resulting in high Infant Mortality Rate (IMR), with these all become even more relevant alongside findings of high Methylmercury exposure.
In addition, we find a suggestion of the double burden of malnutrition postulated in this paper, again not seen in this Munduruku community and largely unexplored in Brazilian indigenous communities, with indigenous communities of Guatemala and Peru showing similar findings, further discussion is found in our paper Line 572. (ref. 107: PS Tallman et al. 2021, The “Double Burden of Malnutrition” in the Amazon: dietary change and drastic increases in obesity and anemia over 40 years among the Awajún;
Ref 108; Ramirez-Zea M et al.2014 The double burden of malnutrition in indigenous and nonindigenous Guatemalan populations. Am J Clin Nutr. 2014 Dec;100(6):1644S-51S. doi: 10.3945/ajcn.114.083857. Epub 2014 Oct 29. PMID: 25411307.)
An Infant mortality rate is recorded through personal interviews for the first time in this community, which have suggested higher levels of infant death than previously thought. Previous studies, which we refer to in our paper (Lima et al 2020.) have used databases to explore indigenous IMR which have acknowledged high rates in the Munduruku community, however, as well as suggesting a higher rate than previously thought, our study helps to identify and describe particular Munduruku communities that are contributing to these high rates.
Overall, for those invested in Brazilian health it offers up a new region exposed to health inequalities. With Brazil still not active within the Minamata convention it is vital to keep contributing to the pool of evidence around unsafe mercury exposure. This paper helps local (Special department for indigenous health) and national (Unified Health System - SUS) health services to identify at risk populations and improve resources to these areas.
For those less closely aligned with health research in amazon, we present levels of exposure recognized globally as unsafe and harmful to developing fetuses. With a very real prospect of future generations of these communities suffering from the harmful effects of toxicity whilst being restricted in obvious neglect and poverty, not conducive with the SDG’s 2+3.

Reviewer 2 Report
Review for IJERPH July 2021
Authors: The authors have done a study of 3 semi-isolated communities in regions associated with artisanal small scale gold mining. The study clearly shows that these communities have exposure to mercury and that is important to know for public health authorities and governmental agencies. The study would have been strengthened by including a community not associated with gold mining since the implication is that the gold mining is influencing the exposure. However, the study itself is quite interesting and worthy of publication.
The study has some significant limitations, some of which the authors acknowledge. The cohort is small, the DDST is a screening test and as such has limited accuracy, and control of covariates is challenging in such isolated communities.
Small studies such as this, especially in environments where one cannot control for the numerous covariates that are already known to adversely influence child neurodevelopment cannot be scientifically used to determine causal relationships.
The authors interpret the threshold dosage as a bright line even though they comment (p3 section 2.3) that it is a safe daily dosage. Reference dosages are not bright lines. Additionally, most have a safety factor included (it is 10 for the US EPA RfD, or 1/10th of the lowest level reported by anyone to be associated with harm). Most are also safe doses to consume daily for a lifetime. Using them to define harm has no scientific basis.
I encourage the authors to rewrite the paper in a factual scientific manner with what they actually studied and to avoid the editorializing, selective quoting of literature, and efforts to establish causality. Larger studies have not found any consistent association of infant anthropometrics with mercury exposure. The levels of exposure are really not alarming. Populations all over the world including in the US (Fletcher 2013 J Community Health) have exposure of this magnitude with no clear adverse health effects.
Author Response
Dear Editor in chief and Dear Reviewer 2
We hope this message finds you well.
Concerning the review report form about the manuscript “Chronic methylmercury exposure in Munduruku indigenous women of childbearing age and its consequences for their children under 2 years old - ijerph-1283940”, we would like to thank you for the valuable technical support and the questions pointed by the reviewers.
Following your suggestions, we promote a comprehensive revision in the paper, including changes in title, general objective, results, and discussion. Please see below, the point-to-point response to the reviewer’s comments.
We try to accommodate all reviewers' questions in order to reach an innovative approach and perform a deeper analysis. Moreover, at the end of the paper, we included a piece of suggestions to the residents on how to reduce health risks.
We hope that our manuscript has enhanced quality and is now more suitable to publish in the International Journal of Environmental Research and Public Health (IJERPH).
Please, let us know if you have additional questions.
Kind regards
Comments and Suggestions for Authors
Review for IJERPH July 2021
Authors: The authors have done a study of 3 semi-isolated communities in regions associated with artisanal small scale gold mining. The study clearly shows that these communities have exposure to mercury and that is important to know for public health authorities and governmental agencies. The study would have been strengthened by including a community not associated with gold mining since the implication is that the gold mining is influencing the exposure. However, the study itself is quite interesting and worthy of publication.
Reply: We agree a control would be useful, unfortunately this is difficult within a region so heavily exposed to ASGM. It is something that we will take into account in future studies. We refer in our paper to a recent study showing a correlation between geographical location to ASGM sites and increased Mercury levels (Vega et al. 2018), whilst we also refer to the difficulty in identifying control groups (Santos-Sacremento et al 2021). Moreover, as mentioned in the recently published paper of our research team, the Pariri Indigenous Association (representing the Munduruku indigenous people living in the Middle-Tapajós Region) sent a letter to the Oswaldo Cruz Foundation to request an assessment of mercury contamination in those three specific communities (Sawre Muybu, Sawre Aboy and Poxo Muybu) These communities have been suffered long-lasting impacts of ASGM for several years. In response, a multidisciplinary, specialized team was formed to put together a comprehensive plan of action and then deliver the fieldwork. In this paper, we devoted particular attention to infants and women of childbearing age.
We have adapted our paper to create a study presenting broader health indicators which we feel now provides a more novel, more informative and generally more useful contribution for the Munduruku population, relevant health services and the wider scientific community.
We have focused on presenting a comprehensive analysis of the health status of this community using MeHg exposure, malnutrition, vaccine coverage and maternal health indicators. Our findings show clear vulnerabilities and urgent action.
There exists one other paper with a similar method published in 2013 (Coimbra et al. 2013, referenced in our paper), however given the isolation, variety of cultural practices and heterogeneity in healthcare provision and infrastructure of indigenous communities along all Brazilian territory, studies of this nature are far more relevant, informative and much needed on a smaller scale. The innovation of our paper lies in the intimate focus of one indigenous community regarding risks to development, morbidity and mortality for future generations of Munduruku through MeHg exposure, low vaccine coverage, and undernutrition alongside deficits in health and sanitation services.
We present a study on MeHg exposure in the Brazilian Amazon in collaboration for the first time with British or even European universities. Santos-Sacremento et al 2021 state specifically that international collaboration would enhance dialogue and recognition of this subject within the international scientific community.
Finally, this study was requested by local leaders and supported by the Local department of indigenous health showing the heightened vulnerable state they are in and the necessity of this study. This is alongside the very real current political vulnerability indigenous communities are facing (Crespo Lopez et al. 2021).
This community has never before been analysed and it is important to continue to map the burden of mercury exposure within this region (Santos-Sacremento et al. 2021).
The study has some significant limitations, some of which the authors acknowledge. The cohort is small, the DDST is a screening test and as such has limited accuracy, and control of covariates is challenging in such isolated communities.
Reply: Thank you, we do acknowledge the limitations and have suggested that there exists at least less bias when examining gross motor skill in comparison to language for example.
Small studies such as this, especially in environments where one cannot control for the numerous covariates that are already known to adversely influence child neurodevelopment cannot be scientifically used to determine causal relationships.
Reply: This is understood and we have further clarified this in our limitations. We have reduced our discussion and results regarding any causal relationships and only continue to discuss the association between maternal and infant hair mercury. Finally, our team could not collect longitudinal data to assess possible changes in the health situation over time (for example, related to seasonality, typical in the Amazon region). Given these limitations, it is impossible to make more robust causal inferences about mercury exposure in the region.
The authors interpret the threshold dosage as a bright line even though they comment (p3 section 2.3) that it is a safe daily dosage. Reference dosages are not bright lines. Additionally, most have a safety factor included (it is 10 for the US EPA RfD, or 1/10th of the lowest level reported by anyone to be associated with harm). Most are also safe doses to consume daily for a lifetime. Using them to define harm has no scientific basis.
Reply: I hope we have cleared up any misunderstandings surrounding reference dosages in our introduction. We do understand the concept and have introduced a higher and more locally relevant reference dose, which we have justified in the introduction. Moreover, to clarify this question, the reference level of 6.0µg/g was used only to identify infants and women of childbearing age with increased exposure to the MeHg, suggesting that these vulnerable groups are at high risk of developing health harms.
Please see below the revised text:
“There is little (or no) consensus amongst agencies in their recommendations for the safe ingestion of methylmercury. In 1972, the Joint Food and Additive Organization / World Health Organization (FAO/WHO) Expert Committee on Food Additives (JECFA) established a provisional tolerable weekly intake (PTWI) for methylmercury equal to 3.3µg/kg bw/week, drawing on health endpoints from poisoning episodes in Minamata and Niigata in the 1950s [44]. Years later, in 1997, United States Environmen-tal Protection Agency (U.S.EPA) proposed a Reference Dose (RfD) for methylmercury of 0.1 µg/kg bw/day based on the intoxication tragedy in Iraq, which was revised and maintained after the Faroes Island cohort study [45]. More recently, in 2003, the JEFCA established PTWI for most vulnerable groups, as women of childbearing age and chil-dren, of 1.6 µg/kg bw/week, and for adults in general of 3.2 µg/kg bw/week [46].
These safe intake doses proposed by different agencies correlate with mercury ex-posure biomarkers, such as blood and hair, and the levels detected in these matrices can be used as a risk exposure indicator. For example, in 1989, the JEFCA proposed a PTWI of 0.3 mg of total mercury per person of which cannot surpass 0.2 mg of methylmercury [47]. This PTWI was converted to hair mercury levels of 6.0 µg/g for identify-ing individuals with high mercury exposure in the New Zealand cohort [48]. Many studies developed in the Amazon region used this mercury level as a reference for the appearance of health effects [49–51], including research with indigenous peoples with a history of or suspected exposure to mercury used in illegal gold mining [13,52]”.
I encourage the authors to rewrite the paper in a factual scientific manner with what they actually studied and to avoid the editorializing, selective quoting of literature, and efforts to establish causality. Larger studies have not found any consistent association of infant anthropometrics with mercury exposure. The levels of exposure are really not alarming. Populations all over the world including in the US (Fletcher 2013 J Community Health) have exposure of this magnitude with no clear adverse health effects.
Reply: We have acknowledged your comments and we hope you feel that we have improved our script. We have paid close attention to the above referenced paper, Fletcher 2013, whilst also exploring the many papers cited within Fletcher 2013 detailing the burden of mercury exposure within New York and the US – We reference this in subsection 4.1 H-Hg in women of childbearing age, in the discussion section. Our results display hair Mercury levels that with the conversion to blood levels are showing a 100% prevalence above the safety factor of 10 dosages that Fletcher refers to of 5.8ug/L. We have aimed to clear up any misunderstandings by adapting our reference limits. Our reference level and the justification can be found in our introduction. In the context of BMDL’s found elsewhere in literature1 (35-58µg/L Blood mercury level translates to 7.7-12µg/g Hair mercury content) showing a prevalence in this population above 40% and 15%, respectively, whilst the average hair mercury concentration of women in one of the communities is 12.6ug/g. The levels in our study are concerning and are showing chronic exposure close to international BMDL’s. Our chosen reference number is not dissimilar to this lower BMDL but we explain and justify the importance of using of 6.0ug/g in our introduction. Finally, we have discussed the context of these levels globally within our discussion with reference to a recent review of mercury levels in 72 countries, further confirming exposure levels in this studied population, even amongst high frequency fish consumers, to be some of the highest in the world (Basu et al. 2018).
Papers referenced in Fletcher 2013 using these levels include Hightower et al. 2006 and Mahaffey 2009.

Round 2
Reviewer 1 Report
The article is revised to meet the requirements for publication and is recommended for acceptance.